# Treatment of Wastewater from a Food and Beverage Industry Using Conventional Wastewater Treatment Integrated with Membrane Bioreactor System: A Pilot-Scale Case Study

**DOI:** 10.3390/membranes11060456

**Published:** 2021-06-21

**Authors:** Sabrina Ng Muhamad Ng, Syazwani Idrus, Amimul Ahsan, Tuan Nurfarhana Tuan Mohd Marzuki, Siti Baizura Mahat

**Affiliations:** 1Department of Civil Engineering, Faculty of Engineering, Universiti Putra Malaysia, Serdang 43400, Malaysia; sabng@icloud.com (S.N.M.N.); t_nurfarhana@yahoo.com (T.N.T.M.M.); 2Cheme Advance Services Sdn. Bhd., Wisma Albe Advance, No 2, Jalan BS7/16, Kawasan Perindustrian Bukit Serdang, Seri Kembangan 43300, Malaysia; 3Department of Civil and Environmental Engineering, Islamic University of Technology (IUT), Gazipur 1704, Bangladesh; ashikcivil@yahoo.com; 4Department of Civil and Construction Engineering, Swinburne University of Technology, Melbourne, VIC 3122, Australia; 5Department of Chemical and Environmental Engineering, Faculty of Engineering, Universiti Putra Malaysia, Serdang 43400, Malaysia; melzura85@gmail.com

**Keywords:** food and beverage wastewater, wastewater treatment plant, membrane bioreactor, hollow fiber, flat sheet, mixed liquor suspended solid and dissolved air floatation

## Abstract

This study compares the performance of the Hollow Fiber (HF) and Flat Sheet (FS) types of membrane bioreactors (MBRs) for the treatment of food and beverage (F&B) industry wastewater in a pilot-scale study of a wastewater treatment plant (WWTP). HF and FS membrane configurations were evaluated at two different Mixed Liquor Suspended Solid (MLSS) levels: 6000 mg/L and 12,000 mg/L. The performance of each configuration was evaluated in terms of Chemical Oxygen Demand (COD) and Total Suspended Solid (TSS) removals for effluent quality measurement. The transmembrane pressure (TMP), flux rate, and silt density index (SDI) were monitored and calculated for membrane fouling assessment. The results show that the rejection rates of COD and TSS for HF and FS membrane types were more than 84% for the two different MLSS levels. During the study, the HF membrane recorded 0.3 bar transmembrane pressure, which complies with the recommended range (i.e., two to three times of chemical cleaning). On the other hand, the FS membrane operates without chemical cleaning, and the TMP value was below the recommended range at 0.2 bar. It was found that the flux values recorded for both the HF and FS systems were within the recommended range of 40 L/m^2^/h. Analysis of SDI revealed that the calculated index ranged between 1 and 2.38 and was within the allowable limit of 3. Both types of MBR consistently achieved an 80% to 95% rejection rate of COD and TSS. Effluent quality measurement of treated F&B wastewater in this pilot-scale study using a WWTP integrated with an MBR indicated a good achievement with compliance with the Malaysia industrial effluent discharge standards.

## 1. Introduction

In general, a large amount of water is used for the beverage production, washing, and cleaning process, and approximately 42% of wastewater is discharged to the drains [1]. The increase in demand of food and beverage (F&B) industries in Malaysia results in an increment of the productions line, which leads to a higher volume of wastewater produced. Industrial wastewaters need to undergo a treatment system, which is known as an Industrial Effluent Treatment System (IETS) or Wastewater Treatment System (WWTP), before they can be discharged into public drains or water bodies. Food and beverage industries have been reported to discharge wastewater with a high concentration of organic and inorganic substances. In Malaysia, the Department of Environment (DOE) has emphasized the importance of discharged effluent to comply with the discharged standard as addressed in the Environmental Quality Act (EQA) 1974. Failure to comply with this standard will cause the factory to be fined by the authorities and the plant to be sealed, which leads to severe losses due to production shutdown. The major concern is the limitation of the footprint on the existing wastewater treatment plant, which creates an opportunity for the system to be integrated with a membrane bioreactor (MBR). This study will specifically focus on wastewater treatment in the F&B industry with the application of an MBR at pilot scale. The MBR system consumes high capital cost and has high energy consumption [2]. This provides critical challenges due to the higher investment needed to build up a wastewater treatment plant, especially in Malaysia. On top of that, the major maintenance of an MBR involving the replacement of membrane elements and a permeate adaptor will require an additional cost, which may be the drawbacks of MBR system [3].

In general, a WWTP consists of physical chemical treatment or biological treatment; a large amount of water is used for the beverage production, washing, and cleaning process, and approximately 42% of water generated was discharge into the drain. The owner of the premises is required to construct a WWTP to collect the processed water and treat the wastewater generated within their premises, and they must comply with the specification as specified in the Guidance Document on the Design and Operation of IETS issued by the DOE [4]. The wastewater from the F&B industry mainly contains a high amount of sugar, flavorings, and coloring additives, which indirectly contribute to the spike of biochemical oxygen demand (BOD) and chemical oxygen demand (COD) in the discharged effluent [5]. Since about 63% of COD is unable to be treated by using a physical process, further treatment by using chemical and biological processes is needed to further reduce the pollutant content in the discharge effluent [1].

In a typical WWTP, effluent flows into an equalization tank (EQ) and is mixed by using a mixer or air blower before it flows into a chemical reaction tank (CRT) where the process of coagulant and flocculation will take place. Following this process, wastewater will overflow to the primary clarifier or dissolved air flotation (DAF) for the solid–liquid separation process. At this stage, clarified water is discharged into biological treatment in an activated sludge reactor (ASR), and the separated solid will be transferred to a holding tank for sludge treatment. An MBR system enables the advantages to be integrated into the existing ASR tank without any or a smaller additional footprint of new tank [6,7]. This is due to the bacteria population (commonly measured in the value of MLSS) cultivated in the ASR tank, which typically falls around 3000 mg/L, whereas after integrating the MBR system into the ASR tank, the bacteria population is capable of reaching up to ≈9000 mg/L [8,9]. Mixed liquor suspended solid (MLSS) is the concentration of suspended solid in the activated sludge reactor that is often used as an indicator for the growth of the microorganism. For healthy growth of the microorganism, it is recommended to maintain the level of dissolved oxygen (DO) at a higher range. Since in this study, the MLSS level is controlled at a concentration between 6000 and 12,000 mg/L, it is important to ensure that the air supply to the system is sufficient to accommodate the growth of microorganisms.

MBR is a type of filtration system that is fully submerged inside the conventional type of activated sludge reactor (ASR) tank. MBR is well known for its effectiveness in the removal of organic and inorganic substances in wastewater [10,11]. Since the scenario in the industry wastewater demand has a limitation regarding the footprint, the MBR system needs to be able to offer the ability to treat high flow rates of water and high loading rates intake with a smaller system footprint [12,13]. In addition, the MBR system has to be capable of providing stable system operation, which may ease the industrial personnel system handling despite its capability to undergo a nitrification and denitrification process in the degradation of organic matter in a wastewater treatment system [14]. MBR is also flexible in terms of its suitability to be installed after undergoing an aerobic or anaerobic pre-treatment system in a wastewater treatment plant [15,16]. Other than producing a better quality of effluent and complied to standard of treated water, the wastewater treatment system generated less sludge production, which led to the significant reduction of sludge disposal cost [3].

The application of an MBR system toward the existing WWTP system is not limited to an improvement of plant capacity and the loading uptake to the system, but the system also tends to develop frequent fouling if it does not undergo proper maintenance [17,18,19,20]. Research shows that under subcritical flux conditions, membrane fouling can exist in three stages, where it will slowly increase linearly and increase exponentially with a rapid liner spike at the end of the process [20]. In general, fouling development is governed by several factors including influent wastewater characteristic, biomass or MLSS concentration, and the operational conditions, which include dissolved oxygen level, temperature, TMP value, sludge retention time, cake layer thickness, and membrane porosity [9,21].

The transmembrane pressure (TMP), flux rate, and silt density index (SDI) are the most commonly used indicators of membrane fouling. TMP is measured by monitoring the pressure of the MBR membrane before operation and during operation by using a pressure sensor [22]. Generally, MBR is a type of filtration system where clean water will pass through the surface of the membrane wall and leave the particulates outside or attached to the membrane wall, which is often known as permeate water [18]. Therefore, the accumulation of the biomass particulates will contribute to the pressure build up over time due to the resistance of water permeating through the membrane wall, which can be triggered by the reduction of flux and the increase of TMP in the system [19,23,24].

Flux can be defined as a mass or volume unit moving or transfer through the membrane surface unit. In a study conducted using confocal laser scanning microscopy (CLSM), at the lowest flux operation, the bio cake accumulation was shown to be higher than the biological substances in the system, which contributed to the increase of permeability resistance during the operation [23]. However, membrane fouling can be recovered by the cleaning of the membrane, whether it is by physical cleaning or chemical cleaning. Studies conducted on the fouling of an MBR membrane testing different MBR configurations such as hollow fiber (HF) type and flat sheet (FS) type membranes shows that there is the formation of a bio cake layer and pore block when the TMP level can reach 50 kPa or 0.5 bar [24].

The membrane configuration is differentiated into a few types; those commonly used for MBR modules are HF and FS. A set of HF elements is assembled to produce a skid of MBR modules. The material of construction for HF and FS elements is polyvinylidene fluoride (PVDF) due to its resistivity toward high chemical concentration. For studies conducted on the application of an MBR system using an HF membrane, the TMP increased to 40 kPa or 0.4 bar within 5 days and 60 kPa or 0.6 bar within 10 days [25]. Furthermore, previous research reported that an HF membrane prone is more to fouling issues with the bio-cake layer deposited at the wall of the membrane [24]. The clean water filter from the FS membrane shows that the percentage of pollutant reduction is higher, which is at the range of 97% to 99% as compared to the HF membrane [24]. The chemical cleaning for an FS membrane is recommended by the manufacturer to perform cleaning with the frequency of once every six months, whereas an HF membrane needs to undergo weekly cleaning [26]. The application of HF membrane configuration is widely used in Malaysia wastewater treatment plants, especially in the F&B industry. The study comparison between HF and FS membranes shall be highly recommended to influence the industry buyer to explore the choice of technology that the system may offer to solve the footprint issue.

In some F&B industries, an MBR system was introduced at the pre-treatment stage together with an oil skimmer and the introduction of ultraviolet (UV) pre-disinfection for the reduction of suspended solids and a huge amount of dissolved organic impurities, approximately above 99%, before entering a double stage of nanofiltration [27]. On top of that, studies conducted on the combination of packed bio-balls with an MBR system state that the results are stable in terms of the removal of organic constituents where there was an average of 94% reduction of the COD and Total Organic Compound (TOC) by differing the hydraulic retention times during the experiment [28]. Moreover, studies at the laboratory scale for the treatment of soft drink industry wastewater by using an MBR system show a combination of anaerobic and aerobic systems where an anaerobic Expanded Granular Sludge Bed (EGSB) as pre-treatment enters the aerobic tank before undergoing the MBR system; these studies reported that the COD removal efficiency ranged from 60 to 87% [29]. In general, F&B industrial wastewater treatment plants integrate the MBR system in the biological treatment. Although different treatment processes produce different final effluent characteristics, the removal efficiency of the pollutant is high. A previous study shows that the COD reduction after the MBR system can be reduced up to 83.9% and above, whereas the TSS reduction is above 93%. The MBR system application in the treatment of F&B industry wastewater is shown in Table 1.

Most previous studies focused on small-scale systems with capacity of less than 100 L, and little consideration has been given to the pilot scale with a capacity of 1000 L for performance assessment. Furthermore, none of the previous studies have compared MBR performance by using different membrane configurations of HF and FS at the pilot scale. The aim of the study is to investigate the performance of different MLSS levels of F&B wastewater by employing two different membrane configurations, which include HF and FS types of MBR. Performance analysis in terms of effluent quality as well as fouling index will be addressed in the following section.

## 2. Materials and Methodology

This study specifically focused on an MBR system after it undergoes the chemical treatment process in a pilot-scale WWTP with a design capacity of 1000 L/d which is located in Nilai, Negeri Sembilan, Malaysia as shown in Figure 1a. The samples were collected and analyzed according to parameters including pH, COD, and TSS. The pilot-scale MBR system was designed to treat the wastewater produced from F&B industry operated by Kian Joo Canpack Sdn Bhd. The effluent flows into the equalization tank (EQ) and is mixed by using a submersible mixer before it flows into a chemical reaction tank (CRT) where the process of coagulant and flocculation will take place. Then, the wastewater will overflow to dissolved air floatation (DAF) for the solid–liquid separation process. At this stage, clarified water is discharged into biological treatment, which includes an activated sludge reactor (ASR) with an integrated MBR system before discharge to the drain, whereas the separated solid will be transferred to a holding tank for sludge treatment. The schematic diagram of a pilot plant MBR system is shown in Figure 1b.

### 2.1. Sample Collection

The wastewater sample was collected at 3 different sampling points: the raw sample located at the influent of the EQ, the effluent from the DAF system, and the effluent of the MBR system. The samples were analyzed using different laboratory instruments to measure the parameters required such as pH, COD, TSS, turbidity, and dissolved oxygen. The process flow of study as shown in Figure 2.

### 2.2. Analytical Method (Laboratory Analysis)

The samples were collected from a beverage manufacturing factory located at Nilai, Negeri Sembilan, Malaysia and tested according to the method adopted from the Standard Methods for the examination of water and wastewater analysis [30]. COD was measured using an HACH Spectrophotometer DR6000 (Standard Method 5220 D for low range value and high range value) after being heated in COD reactor at 150 °C for 2 h. TSS measurement was carried out after filtration using a 0.45 µm filter on the vacuum pump, and the sample was heated in a drying oven at 103 °C. The gravimetric method was applied for TSS measurement.

### 2.3. Experimental Design

The samples were collected from different points of the wastewater treatment system and the frequency of sampling and monitoring was 5 days for 3 consecutive weeks, as shown in Table 2. The wastewater flows through 2 types of membrane configuration: HF and FS types of MBR module. The HF membrane was integrated in the ASR, while the FS membrane is integrated in the MBR tank where the effluent of DAF is partially channeled to the MBR tank. A pump was installed in the outlet of each MBR module and equipped with an electromagnetic flowmeter to measure the flow of water discharge from the system. An air blower was supplied air to the air distribution piping located underneath the MBR module, and a pressure transmitter was installed to monitor the static pressure and operation pressure where the reading is used to calculate the TMP of the MBR system. The level of MLSS in the MBR system is controlled at 6000 mg/L and 12,000 mg/L for FS and HS configurations. Membrane cleaning in place (CIP) is recommended by the manufacturer to be performed about once a week for maintenance cleaning or once every 3 months for major recovery cleaning by using sodium hypochlorite (NaOCl) (range from 300 to 500 mg/L) or when the transmembrane pressure exceeds the set value limit. During this study, maintenance cleaning is performed to prevent the increase of transmembrane pressure, which would affect the membrane permeability.

### 2.4. Membrane Configuration

In this study, the wastewater has undergone two different types of MBR membrane configuration. The details specification of each membrane is tabulated in Table 3.

### 2.5. Membrane Fouling

Membrane fouling can be measured and monitored using a few parameters that include transmembrane pressure, membrane flux, and silt density index. TMP was measured from an installed pressure transmitter at the permeate pipeline. The value difference of static pressure and operating pressure was used to calculate the TMP. The reading is recorded as an indicator of membrane fouling [31]. Calculation of the TMP is as follows:(1)$$P_{tm}=\frac{\left(P_{i}+P_{o}\right)}{2}-P_{p}$$
where

*P_tm_* = Transmembrane pressure (bar)

*P_i_* = Pressure at the inlet of the membrane module (bar)

*P_o_* = Pressure at the outlet of the membrane module (bar)

*P_p_* = Permeate pressure (bar).

Membrane flux was calculated to monitor the performance of the membrane module. However, the flux value varies, and it is subjected to the characteristic of wastewater. The flux is expressed in units of L/m^2^/h and usually set at the range of 18.8–30.1 L/m^2^/h [32]. The equation of flux can be expressed as follows:(2)$$J=\frac{V}{A\times T}$$
where

*J* = Flux (L/m^2^/h)

*V* = Water volume in permeate (L)

*A* = Membrane contact area (m^2^)

*T* = Time (h).

Silt density index (SDI), on the other hand, is a test to calculate the relationship of filtration over time. The SDI is calculated by using Equation (3) [33,34,35].
(3)$$SDI=\frac{1-\left(\frac{t_{i}}{t_{f}}\right)}{T}\times 100$$
where

*t_i_* = Initial Filtration time

*t_f_* = Final Filtration time

*T* = Total time during filtration.

Approximately 500 mL of sample is passed through a filter or membrane of 0.45 µm for 15 min duration time. The time of initial and final filtration is recorded in seconds. On top of that, membranes with high rejection indicate good membrane characteristics. The rejection rate is calculated as follows:(4)$$R\;\%=\left(1-\frac{C_{p}}{C_{f}}\right)\times 100$$
where

*R* = Rejection %

$C_{p}$ = Concentration in permeate

$C_{f}$ = Concentration in feed.

## 3. Results

In this section, the percentage removal of each parameter and the overall performance for both configurations and MLSS are discussed. 

### 3.1. Characteristics of Raw F&B Wastewater

Raw F&B wastewater was collected at the influent of EQ and characterized as tabulated in Table 4.

### 3.2. Characteristic and Quality of F&B Effluent in a DAF System

After the coagulation and flocculation process, F&B wastewater is passed through a DAF system where high air pressure is supplied. The floating sludge is removed to a holding tank for sludge management, and the clarified water flows into the ASR for further treatment. Characteristics of the effluent in a DAF system are shown in Table 5.

### 3.3. Characteristic and Quality of F&B Effluent in an MBR System

Partially treated F&B wastewater was further processed for discharged standard compliance. The MBR system integrated in the ASR was employed to further treat F&B wastewater before being discharged into the water bodies or drain. During this stage, two different MLSS were applied, and the results are shown in Table 6.

### 3.4. Continuous Performance Monitoring

COD is a crucial parameter in evaluating the performance of an industrial WWTP. Thus, all WWTP must be designed according to the desirable range of raw influent COD to ensure the effluent of the WWTP complied to the discharge standard. Figure 3 and Figure 4 show that the COD influent and effluent obtained for an MBR system are slightly stable with a high COD percentage of removal efficiency, which complies with discharge standard A and standard B.

### 3.5. TSS Analysis

In a wastewater treatment plant, the measurement of TSS is crucial to ensure that solids are removed from the wastewater before it is discharged into water bodies. Consistency in the monitoring of TSS will ensure that the WWTP operates at optimum performance and eliminates the possibility of system failure, hence reducing costs for sludge management. Figure 5 and Figure 6 shows that the incoming TSS concentration was to further improve with a high percentage of TSS removal efficiency after passing to the MBR, complying with Standard A.

### 3.6. Transmembrane Pressure (TMP) Analysis

TMP requires frequent monitoring, as this parameter is one of the indicators of performance of an MBR system. This is due to the decrease of membrane permeability in operation, which was reported to be the major drawback to the MBR technology and mainly contributes to the membrane fouling issue [32,36]. Figure 7 shows the TMP monitoring for HF during its operation at two different concentrations of MLSS.

### 3.7. Flux Analysis

Flux is another measuring parameter during the monitoring of membrane fouling. Operating at higher design fluxes can inevitably increase operating costs due to higher operating pressures, more frequent cleaning, and potential membrane replacement costs. Figure 8 shows the flux monitoring at two different concentrations of MLSS.

### 3.8. Silt Density Index (SDI)

In addition to the analysis of TMP and flux, membrane fouling can also be measured using the silt density index (SDI), which is also known as the fouling index, where two different samples are taken and a test is conducted, and the results are shown in Table 7.

## 4. Discussion

The results in Table 4 show that raw beverage manufacturing wastewater has a neutral range of pH. COD was found to be 50% higher in a tank with an FS membrane, which is 3000 mg/L as compared to that with an HF membrane at 1710 mg/L. The TSS value for the incoming wastewater ranges from 140 to 250 mg/L. A previous study reported that the influent wastewater from beverage processing wastewater contained approximately 590–1350 mg/L of COD and 77–120 mg/L of SS before further treatment [19]. Moreover, the feed COD may increase up to 9950 mg/L for the soft drink industry due to the sugar content in certain products [29]. Dairy and soy beverage products showed significantly high COD content of up to 9500 mg/L and 12,000 mg/L respectively [5].

The obtained data in Table 5 show a reduction of COD concentration for HF and FS membranes of 87% and 49% respectively. A similar trend was recorded for TSS concentration with 49% and 62% reduction for the HF membrane and FS membrane, respectively. A previous study of dairy wastewater treatment found that the removal percentage of TSS and COD can be reduced up to 91% and 50% respectively by using a combination of coagulant and DAF [37]. Another study conducted using the same method showed that the TSS and COD reduction can be up to 77.5% and 88.7%, respectively [38]. In addition, previous research has found that DAF treatment reduces the dissolved solids by 20% and 90% for suspended solids [39].

In addition, Table 6 shows a comparison of different MLSS for HF membranes indicating a significant removal of both COD and TSS between 85% and 95%. This indicates that at a difference of 6 g/L of MLSS, the performance of the HF membrane was not very affected. The findings for the FS membrane are also in line with those of the HF in terms of different MLSS applied. Nevertheless, comparing the performance between different configurations, the concentration of COD in the final effluent was found to be 86% higher in FS as compared to HF, where the final effluent is 128 mg/L and does not comply with the discharged standard. This indicates more effective F&B wastewater treatment as compared to the HF system.

Overall, the HF membrane recorded the highest reduction for both parameters and the HF membrane shows the best performance for the MLSS value of 6000 mg/L with COD and TS reduction recorded at 92.6% and 94.7%, respectively. However, the figure shows a slight drop when the MLSS value increased to 12,000 mg/L, where the COD reduction was 85.5% and the value of TSS reduction was maintained at 94.1%. This finding is in line with a previous study conducted using a ceramic type of MBR membrane which reported that at an MLSS of 12,000 mg/L, COD removal rates were more than 95% [26]. On the other hand, the FS membrane for the MLSS values of 6000 mg/L and 12,000 mg/L shows COD and TSS reductions of 91.6% and 84.4%, respectively. The COD concentration was able to be reduced from an average of ≈1537.9 to ≈128 mg/L, whereas the TSS concentration reduced from ≈93.5 to ≈14.0 mg/L after undergoing the MBR system.

From the results, it can be concluded that treated F&B wastewater using an HF membrane can comply with discharged Standard A. However, the FS membrane complies with discharged Standard B only. A previous study conducted using an HF membrane stated that the reduction of organic constituents inclusive of the COD can be captured at an average of 94% with the range of MLSS level between 2790 and 5480 mg/L [28]. Researchers also suggested that the application of an MBR system is suitable as a secondary treatment especially for COD removal and TSS removal, for which the results can be up to 99% and 93.1%, respectively [5].

During the monitoring period for the FS membrane, the incoming COD was higher, which ranged from 1000 to 3000 mg/L. In Figure 4, after passing through MBR, the effluent is only capable of complying with Standard B, Environmental Quality (Industrial Effluents) Regulations 2009 but with high COD removal, as reported by previous study [40].

Figure 6 shows that the data obtained for TSS effluent DAF range slightly higher than the value of TSS during the study operation of the HF membrane within the range of 40 to 224 mg/L of TSS. However, after passing through the MBR system, the treated water results for TSS were recorded within the allowable limit of discharge Standards A and B, which is approximately below 50 mg/L. This finding can be supported by previous research work which described that the MBR system is capable of achieving TSS removal efficiency of almost more than 90% [41].

In addition, Figure 7 shows the TMP trend for the FS membrane at MLSS levels of 6000 mg/L and 12,000 mg/L. The recommended initial TMP by the manufacturer for the MBR system is more than 0.15 bar. By comparing the results in the figure, it is clearly projected that the HF membrane is operating at a high range of TMP and it is close to the recommended range of TMP, which is less than 0.3 bar. On the other hand, there are few readings that dramatically fall to 0 bar. This is because during the operation of MLSS 6000 mg/L, the system required chemical cleaning three times with the frequency interval of once every 3–5 days, whereas at MLSS 12,000 mg/L, the system performed chemical cleaning twice within 4–8 days. After conducting membrane cleaning maintenance with cleaning in place (CIP), the TMP shows improvement and significantly depreciates compared with the TMP before CIP. This is a good indicator for fouling development in the system, which caused the resistance toward MBR performance due to clogged pores and suction pressure increase during the operation [32,42]. On the other hand, the results obtained were relatively low compared to those of the HF type membrane. TMP during the operation of MLSS at 6000 mg/L shows a slight fluctuation as compared to at 12,000 mg/L.

Figure 8 shows that the operating flux for the HF membrane is between 10 and 40 L/m^2^/h. Some of the operating flux is higher than the recommended flux by the manufacturer. Other than that, the operating flux for the FS membrane is in between the recommended flux, which is less than 15 L/m^2^/h. Ideally, the membrane flux usually operates at the lower value to minimize the generation of membrane fouling in MBR [32].

SDI is known as an indicator of the relationships between particles in water and the fouling development in the filtration system. During this study, two separate samples were taken, and the results reveal that the SDI for sample 1 can be up to 2.38, whereas that for sample 2 was maintained below SDI 2, which is around 1.91, as shown in Table 7. The allowable limit for SDI is 3. The limit has been set to minimize the membrane fouling through the recommended membrane cleaning maintenance schedule and pretreatment process if required [43]. However, the value of SDI varies depending on the water characteristics. There was a study conducted on the SDI for an MBR influent sample, which can reach up to 3.24, indicating that the system has fouling potential [34]. SDI was also measured in organic wastewater, where the results obtained ranged from 1.28 to 4.16 due to the different characteristics of the raw sample [33,44].

## 5. Conclusions

Raw F&B wastewater presents pH and COD values that range from 6.9 to 7.4 and from 710 to 3000 mg/L respectively. The TS concentration in the raw samples ranges from 140 to 250 mg/L. All parameters have shown significant reduction after DAF and MBR with reduction of TS and COD recorded between 80 and 95%.

Running the pilot scale for two different types of membranes has shown a great performance for both systems. The overall performance of the HF and FS membranes both shows significantly high percentages of reduction of COD and TSS, which ranged from 84.4% to 94.1% regardless of the different MLSS levels during operations. However, the study shows that the HF with an MLSS of 6000 mg/L shows the highest percentage of reduction: 92.6% and 94.7% for COD and TSS. Fouling development in the MBR system is measured by the monitoring of TMP and flux pattern analysis. The TMP trend of the HF system shows that the system operates just below the recommended range, which is 0.3 bar. This can be observed when the system operates at higher flux and it exceeds the recommended flux given by the manufacturer. Therefore, it can be observed that after chemical cleaning, the flux operation is reduced slightly and significantly improves the membrane permeability to the system. The FS membrane, on the other hand, shows positive results where the flux and TMP trends stay far below the recommended range.

This is related to the flux obtained where the HF membrane operated at a higher flux, while the FS membrane operated at a lower flux. On top of that, during the operation of these membranes, no chemical cleaning activity has been conducted. The SDI obtained for two different samples shows that the SDI for sample 1 ranges from 2.26 to 2.38 and that for sample 2 ranges from 1.46 to 1.91. The SDI obtained for each sample is below the recommended value, which is less than 3. This indicates that the development of membrane fouling is rapid when using the HF membrane as compared to the FS membrane type.

Therefore, in a long-term operation, this contributes to the increase of maintenance cost to the system because the system requires chemical cleaning throughout its operation. Despite the fouling issue, both membrane configurations have proven capable of further reducing the pollutant content in the wastewater even at different concentrations of MLSS, and the results obtained at the effluent of MBR do comply to the discharge standards in the Environmental Quality (Industrial Effluents) Regulations 2009. Additionally, several recommendations are listed for future work to improve the knowledge of the related study area. More research work should be carried out on F&B wastewater treatment at higher MLSS concentration to investigate the efficiency of the system and the effect on fouling deposition. Similar evaluations can be performed on other types of modules, including spiral wound, at various MLSS levels.

## Figures and Tables

**Figure 1 membranes-11-00456-f001:**
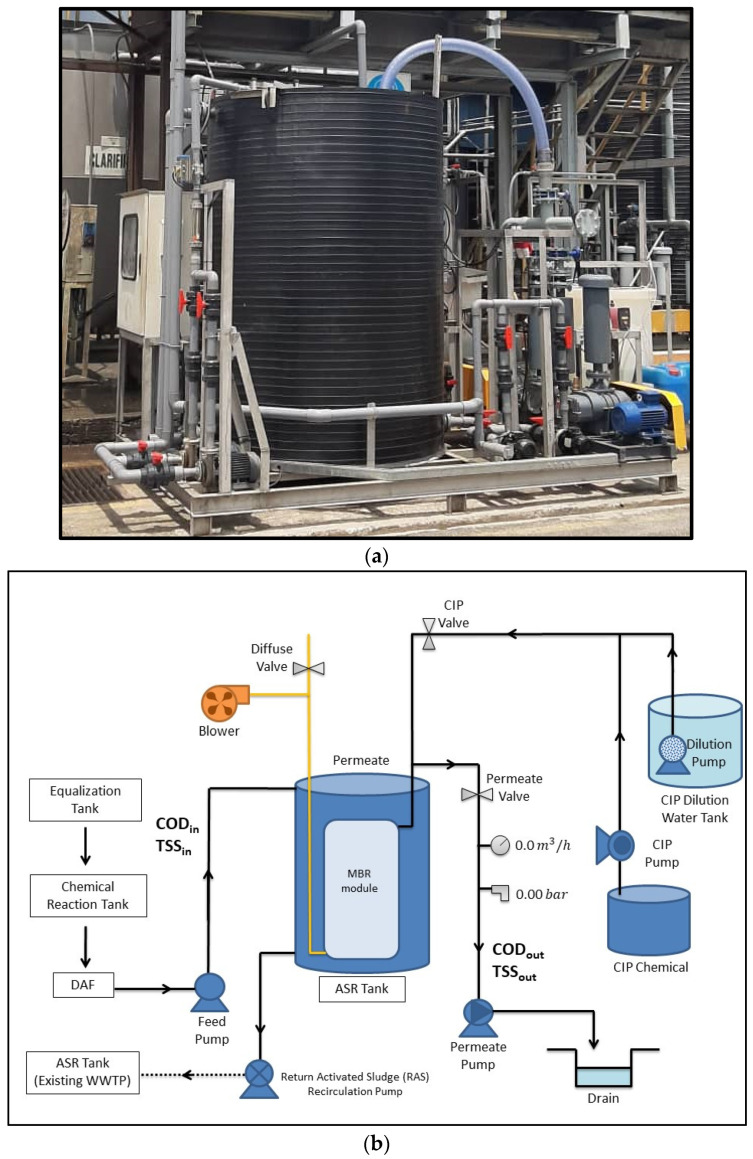
(**a**) The pilot plant MBR system set up (source: Beverage Manufacturing Wastewater Treatment Plant, Cheme Advance Services Sdn Bhd); (**b**) The pilot plant MBR system schematic diagram.

**Figure 2 membranes-11-00456-f002:**
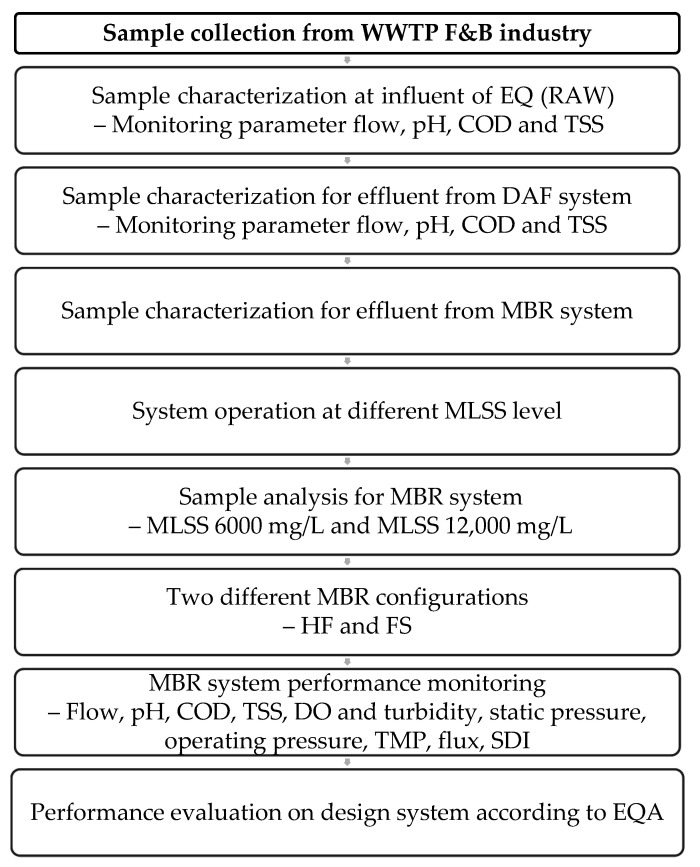
Process flow of study.

**Figure 3 membranes-11-00456-f003:**
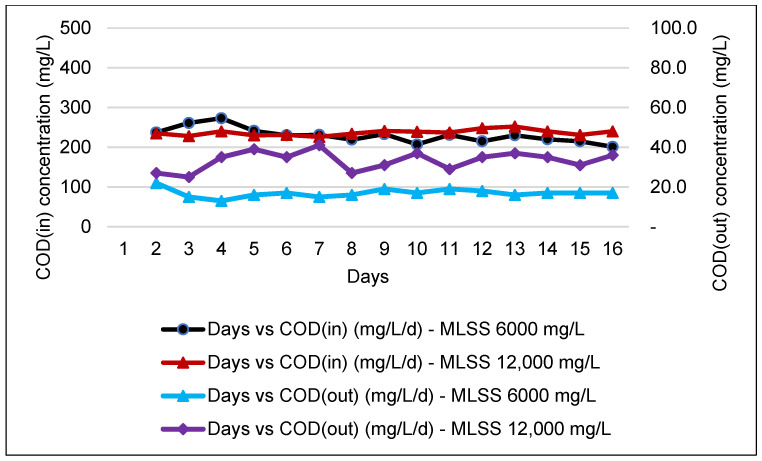
COD concentration of influent and effluent for an MBR system at MLSS values of 6000 mg/L and 12,000 mg/L by using an HF membrane.

**Figure 4 membranes-11-00456-f004:**
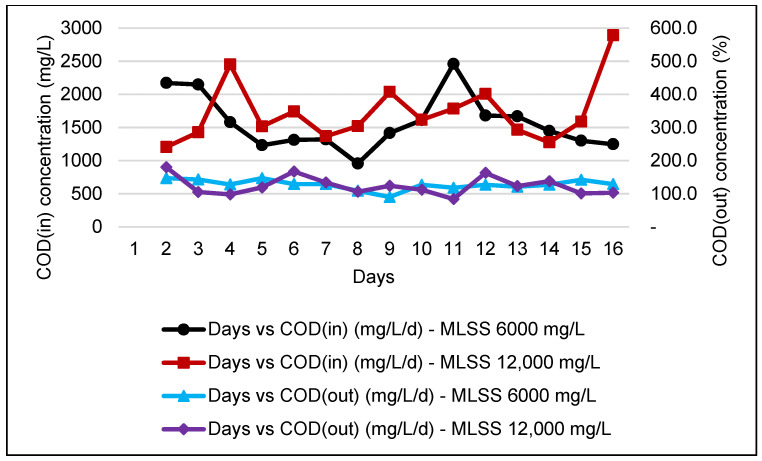
COD concentration of influent and effluent for an MBR system at MLSS values of 6000 mg/L and 12,000 mg/L by using an FS membrane.

**Figure 5 membranes-11-00456-f005:**
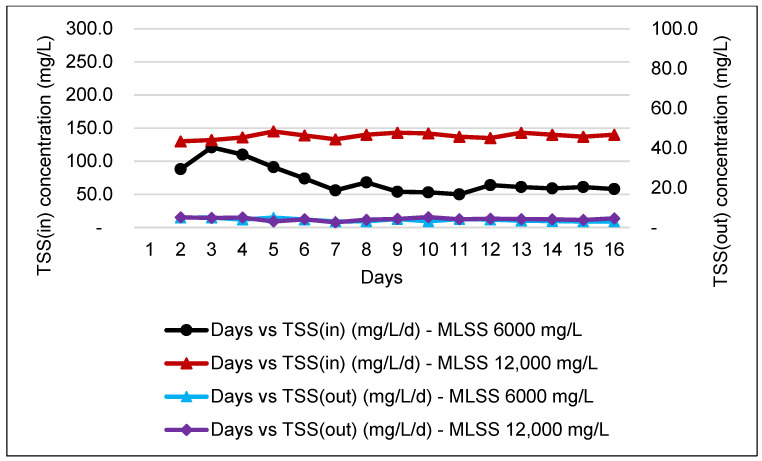
Graph of TSS influent and effluent for an MBR system at MLSS values of 6000 mg/L and 12,000 mg/L by using an HF membrane.

**Figure 6 membranes-11-00456-f006:**
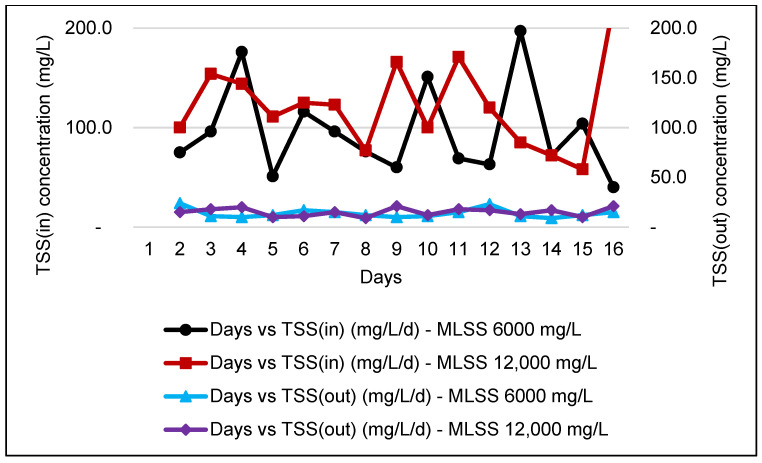
Graph of TSS influent and effluent for an MBR system MLSS values of 6000 mg/L and 12,000 mg/L by using an FS membrane.

**Figure 7 membranes-11-00456-f007:**
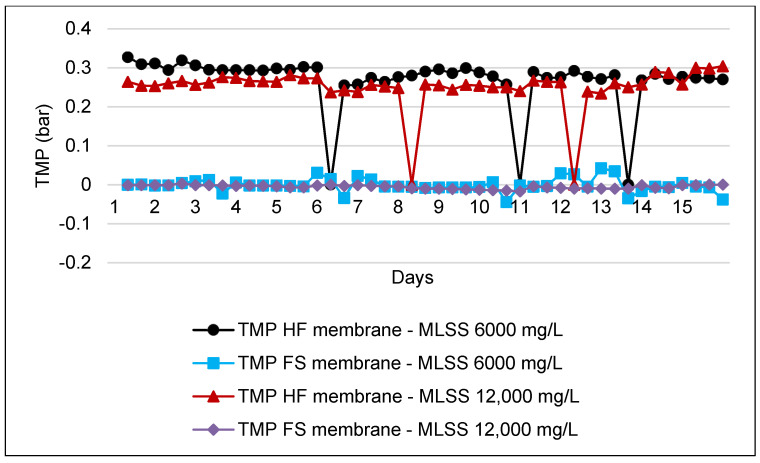
TMP trend for HF and FS membranes by using MLSS values of 6000 mg/L and 12,000 mg/L.

**Figure 8 membranes-11-00456-f008:**
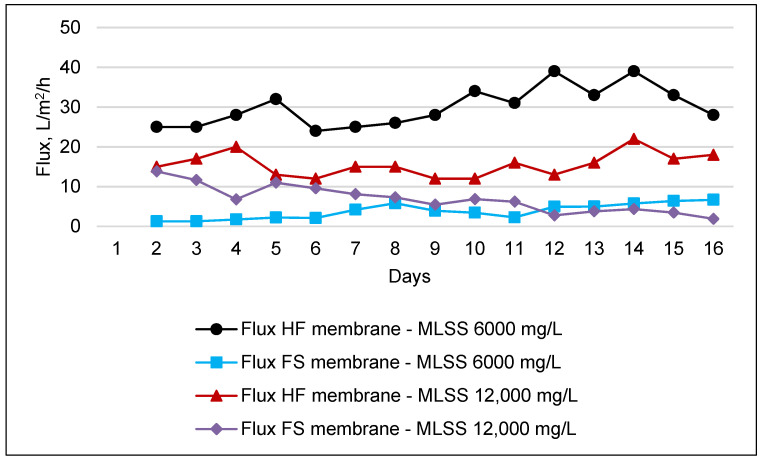
Flux trend for HF and FS membranes by using MLSS values of 6000 mg/L and 12,000 mg/L.

**Table 1 membranes-11-00456-t001:** The summary of MBR treatment for food and beverage industry wastewater.

Source Wastewater	Membrane Type	Pore Size (µm)	Membrane Surface Area (m^2^)	Capacity (L)	Removal Efficiency (%)	Country	Reference
**F&B** **processing plant**	HF	0.04	74	1500	≈99% COD	Europe	[26]
**Beverage** **processing**	HF	0.4	0.92	40	≈94% COD	Croatia	[27]
**F&B** **processing plant**	HF	0.036	0.046	50	≈91–98% COD	USA	[18]
**Soft drink** **processing**	HF	0.2–0.4	0.058	30	≈83.9% COD	South Africa	[28]
**Dairy and soy** **processing**	HF	0.5	0.044	10	≈93.1% TSS ≈99% COD	New Zealand	[3]

**Table 2 membranes-11-00456-t002:** The monitoring parameter and the sampling point.

No	Monitoring Parameter	Point of Sampling
1	Flow	Influent EQ, Effluent DAF, Effluent MBR
2	pH	Influent EQ, Effluent DAF, Effluent MBR
3	COD	Influent EQ, Effluent DAF, Effluent MBR
4	TSS	Influent EQ, Effluent DAF, Effluent MBR
5	Turbidity	Effluent MBR
6	DO	Effluent DAF
7	Static Pressure	Effluent MBR
8	Operating Pressure	Effluent MBR
9	TMP	Effluent MBR
10	Flux	Effluent MBR

**Table 3 membranes-11-00456-t003:** MBR module specification.

No	Specification	Unit	HF	FS
**1**	Brand	-	Sterapore	Membray
**2**	Make	-	Mitsubishi, Japan	Toray, Japan
**3**	Membrane Surface Area	m^2^	1000	45
**4**	Material	-	PVDF	PVDF + PET non-woven fabric
**5**	Pore Size	µ	0.4	0.08
**6**	Recommended MLSS range	mg/L	5000–12,000	7000–18,000
**7**	Recommended operating TMP	bar	<0.3	<0.2
**8**	Recommended operating Flux	L/m^2^/h	<33.3	<31.2
**9**	Air flow rate	Nm^3^/min/module	4.4	0.75

**Table 4 membranes-11-00456-t004:** Raw wastewater characteristics of the influent of an EQ tank for two different membrane configurations.

No	Parameter	Unit	Results	Standard A *	Standard B *
	HF Membrane				
**1**	Flow	m^3^/h	30	-	-
**2**	pH	pH	7.4 ± 1	6.0–9.0	5.5–9.0
**3**	COD	mg/L	1710 ± 312	80	200
**4**	TSS	mg/L	140 ± 65	50	100
	FS Membrane				
**5**	Flow	m^3^/h	25	-	-
**6**	pH	pH	6.9 ± 11	6.0–9.0	5.5–9.0
**7**	COD	mg/L	3000 ± 312	80	200
**8**	TSS	mg/L	250 ± 65	50	100

* Standard A and Standard B are acceptable conditions for the discharge of industrial effluent for mixed effluent extracted from Environmental Quality (Industrial Effluents) Regulations 2009.

**Table 5 membranes-11-00456-t005:** Characteristics of effluent in a DAF system.

No	Parameter	Unit	Average Results	Total Reduction (%) (Influent from EQ Tank)
	**HF Membrane**			
**1**	COD	mg/L	229.6	86.5
**2**	TSS	mg/L	71.2	49.1
	**FS Membrane**			
**3**	COD	mg/L	1537.9	48.7
**4**	TSS	mg/L	93.5	62.6

**Table 6 membranes-11-00456-t006:** Sample characteristics of the effluent in the MBR system.

No	Parameter	Unit	Average Results	Total Rejection Rate (%) (Effluent from DAF)
	**HF Membrane at MLSS 6000 mg/L**			
**1**	COD	mg/L	16.9	92.6
**2**	TSS	mg/L	3.8	94.7
	**HF Membrane at MLSS 12,000 mg/L**			
**3**	COD	mg/L	33.3	85.5
**4**	TSS	mg/L	4.2	94.1
	**FS Membrane at MLSS 6000 mg/L**			
**5**	COD	mg/L	128.0	91.7
**6**	TSS	mg/L	14.5	84.5
	**FS Membrane at MLSS 12,000 mg/L**			
**7**	COD	mg/L	128.7	91.6
**8**	TSS	mg/L	14.6	84.4

**Table 7 membranes-11-00456-t007:** Analysis of SDI.

**HF Membrane Water Sample 1**					
**Time**	Unit	Test 1	Test 2	Test 3	Average
**Time initial, t_i_**	s	330	312	321	321
**Time final, t_f_**	s	212	206	210	209
**SDI**	(%/min)	2.38	2.26	2.31	2.32
**FS Membrane Water Sample 2**					
**Time**	Unit	Test 1	Test 2	Test 3	Average
**Time initial, t_i_**	s	411	423	398	410.7
**Time final, t_f_**	s	298	302	311	304
**SDI**	(%/min)	1.83	1.91	1.46	1.74

## Data Availability

Not applicable.

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
