# Peer review of "Treatment of Wastewater from a Food and Beverage Industry Using Conventional Wastewater Treatment Integrated with Membrane Bioreactor System: A Pilot-Scale Case Study"

_membranes, 2021, doi:10.3390/membranes11060456_

Round 1

Reviewer 1 Report

Please see the file for the comments. 

Author Response

Please find the attached amendment. 

Reviewer 2 Report

The manuscript membranes-1221826 studied the separation performance of Hollow Fiber (HF) and Flat Sheet (FS) membranes in a Membrane Bioreactor (MBR) process to treat the wastewater of a Food and Beverage (F&B) industry. 

The results of this comparative study at pilot-scale are beneficial to membrane literature and I suggest publishing this work after addressing the following comments: 

Major comment:

  • Line 144, Table 1: I recommended the authors to enrich the data by adding more information regarding the properties of feed, properties of the membrane material, operating conditions, fouling performance, etc.   
  • Line 202, Table 5: What’s the rationale behind using HF and FS with large porosity difference, i.e., 0.4 microns and 0.08 microns? What is the authors' justification to have a fair comparison between the outcomes? 
  • Membrane characterization is needed, such as FESEM, Contact angle, FTIR, etc. 
  • Figure 5 and 7: Justification is needed for the larger variation of the influent COD and TSS for FS compared to HF
  • Error bar or standard deviation number is needed for all the data.  

Minor comments: 

  • Line 61: Avoid citing the same reference in a single sentence 
  • Line 70: Define MLSS in the introduction section
  • The manuscript needs improvement in English in several lines. Some examples are
  • Line 91, 98, 107-108, 119-120, 128-131. 
  • Line 145: Table caption should move to the top 
  • Figure 8, and 9: The caption should include both HF and FS

Author Response

Please find the attached amendment.

Round 2

Reviewer 1 Report

My comments can be seen in the attached file.

Author Response

Dear Sir,

Please find the attached for your review.

Reviewer 2 Report

The authors have properly and adequately addressed the comments given on the review process. The revised manuscript is improved in comparison with the original version. Therefore, I recommend this research for publication.

Author Response

Dear Sir,

Please find the attached.
